# Greater Adherence to Dietary Guidelines Associated with Reduced Risk of Cardiovascular Diseases in Chinese Patients with Type 2 Diabetes

**DOI:** 10.3390/nu14091713

**Published:** 2022-04-20

**Authors:** Shang-Ling Wu, Long-Yun Peng, Yu-Ming Chen, Fang-Fang Zeng, Shu-Yu Zhuo, Yan-Bing Li, Wei Lu, Pei-Yan Chen, Yan-Bin Ye

**Affiliations:** 1Department of Nutrition, The First Affiliated Hospital of Sun Yat-sen University, Guangzhou 510080, China; wushling6@mail.sysu.edu.cn (S.-L.W.); zhuoshy@mail.sysu.edu.cn (S.-Y.Z.); luwei2.good@163.com (W.L.); chenpy55@mail.sysu.edu.cn (P.-Y.C.); 2Department of Cardiology, The First Affiliated Hospital of Sun Yat-sen University, Guangzhou 510080, China; penglyun@mail.sysu.edu.cn; 3Guangdong Provincial Key Laboratory of Food, Nutrition and Health, School of Public Health, Sun Yat-sen University, Guangzhou 510080, China; chenyum@mail.sysu.edu.cn; 4Department of Statistics and Epidemiology, School of Public Health, Sun Yat-sen University, Guangzhou 510080, China; 5Department of Epidemiology, School of Medicine, Jinan University, Guangzhou 510632, China; zengffjnu@126.com; 6Department of Endocrinology, The First Affiliated Hospital of Sun Yat-sen University, Guangzhou 510080, China; liyb@mail.sysu.edu.cn

**Keywords:** diet quality score, dietary index, cardiovascular disease, case-control study, type 2 diabetes

## Abstract

The evidence regarding the impact of the scores on healthy eating indices on the risk of cardiovascular events among patients with type 2 diabetes (T2D) is limited. As such, in this study, we examined the associations of adherence to the Chinese and American dietary guidelines and the risk of cardiovascular disease (CVD) among Chinese individuals with T2D. We conducted a 1:1 age- and sex-matched case–control study based on a Chinese population. We used a structured questionnaire and a validated 79-item food-frequency questionnaire to collect general information and dietary intake information, and calculated the Chinese Healthy Eating Index (CHEI) and the Healthy Eating Index-2015 (HEI-2015). As participants, we enrolled a total of 419 pairs of hospital-based CVD cases and controls, all of whom had T2D. We found a significant inverse association between diet quality scores on the CHEI and HEI-2015 and the risk of CVD. The adjusted odds ratios (95% confidence interval) per five-score increment were 0.68 (0.61, 0.76) in the CHEI and 0.60 (0.52, 0.70) in the HEI-2015. In stratified analyses, the protective associations remained significant in the subgroups of sex, BMI, smoking status, tea-drinking, hypertension state, dyslipidemia state, T2D duration, and medical nutrition therapy knowledge (all *p* < 0.05). These findings suggest that a higher CHEI or HEI-2015 score, representing a higher-quality diet relative to the most recent Chinese or American dietary guidelines, was associated with a decreased risk of CVD among Chinese patients with T2D.

## 1. Introduction

Type 2 diabetes (T2D) is a worrying global epidemic that is of particular concern in China [1]. The leading cause of death in T2D remains cardiovascular disease (CVD), and hyperglycemia is associated with increased cardiovascular risk [2]. Patients with T2D have a two- to four-fold higher risk of developing CVD than those without diabetes [3]. As such, identifying cost-effective strategies for the prevention of cardiovascular complications due to diabetes is important.

Apart from pharmacological treatment, dietary modification is a fundamental therapy for self-management of diabetes [4]. Researchers have consistently reported an inverse association between the risk of CVD events and the consumption of individual food items such as fruit [5], vegetables [6], whole grains [7], and seafood [8], and a positive association with unprocessed red/processed meat [9] and salt [10]. Understanding the complexity of multiple dietary exposures and their interrelations, overall dietary pattern analysis, which involves a series methods to assess diets comprehensively, might provide more information about the role of diet in the etiology of diet-related dis-eases than single-food-item analysis [11].

Among these methods, diet quality indices have been developed on the healthy dietary recommendations and available evidence on various diseases to assess compliance with dietary guidelines [12]. Several indices have been developed to evaluate integral dietary quality according to various dietary guidelines, as indicated by e.g., the Healthy Eating Index (HEI), the Mediterranean Diet Quality Index (MDQI), the alternate Healthy Eating Index (aHEI), and the Dietary Guidelines Index (DGI) [13]. Many findings [14,15,16] have shown that higher overall diet scores substantially reduce the risk of cardiovascular events in general populations, but the evidence regarding the effect of diet quality indices following a diabetes diagnosis on the risk of subsequent CVD is limited, particularly in the Chinese population.

Medical nutrition therapy (MNT) plays an important role in the management of diabetes and preventing related CVD complications. Current dietary recommendations focus on promoting healthful eating patterns containing nutrient-dense foods rather than specific nutrients, and on individualized meal planning, emphasizing personal preferences, needs, and goals rather than a generic eating pattern [4]. The Mediterranean, Dietary Approaches to Stop Hypertension (DASH), and plant-based diets are all examples of healthful eating patterns that have shown positive results among T2D patients [17,18,19]. The dietary index calculated from dietary guidelines has wider international recognition than others [20]. Nevertheless, a gap exists in the knowledge of the value of the diet recommended by the dietary guidelines in diabetes management; hence, the clinical practice guidelines for MNT provide no specific recommendations regarding the diet recommended by the dietary guidelines when evidence is lacking [4]. Furthermore, the specific recommendations by the American Diabetes Association (ADA) to reduce the risk of CVD are brief, and no research findings indicate that following these recommendations will decrease an individual’s risk of CVD [21]. Therefore, the dietary patterns that would benefit the prevention of CVD complications of diabetes mellitus need to be explored.

As such, in the current study, we thus investigated the associations between two diet quality indexes, the Chinese Healthy Eating Index (CHEI) [22] and the latest version of the HEI (HEI-2015) [23] (the food-based and food-nutrient-based indices that reflect the 2016 Dietary Guidelines for the Chinese population [24] and the 2015–2020 Dietary Guidelines for the American population [25], respectively), with the risk of CVD among patients with T2D participating in a 1:1 matched case–control study in south China. We provide some additional information for the development of dietary guidelines for the management of T2D.

## 2. Materials and Methods

### 2.1. Study Design

In this case–control study, we included 419 patients with T2D with newly diagnosed CVD and 419 age- (±5 years) and sex-matched T2D-only controls without a diagnosis of CVD who were hospitalized at the Endocrinology Department, the Neurology Department, and the Cardiology Department of The First Affiliated Hospital of Sun Yat-sen University, in Guangdong Province, China, from March 2013 to September 2015. We conducted this study according to the guidelines of the Declaration of Helsinki, and all procedures involving human subjects/patients were approved by the Ethics Committee of The First Affiliated Hospital of Sun Yat-sen University (no. (2017)019, approved on 13 February 2017). Verbal consent was witnessed and formally recorded, and we obtained written informed consent from all patients.

### 2.2. Study Population

#### 2.2.1. Inclusion Criteria

Participants with previously diagnosed T2D, aged between 30 and 85 years, natives of Guangdong Province or had lived in Guangdong for at least 5 years, and with a history of at least 2 years of T2D were considered eligible for inclusion in the study. To conduct a case–control study, we only included patients with an incident (diagnosed within 2 weeks) diagnosis of CVD at a date later than the T2D diagnosis in the case population. The control group included patients with T2D who never had a self-reported CVD incident, exhibited no symptoms of cardiac involvement, had normal EKG levels, and had negative exercise tests.

#### 2.2.2. Exclusion Criteria

We excluded participants with (1) confirmed type-1 diabetes or gestational diabetes mellitus (*n* = 20); (2) previous history of cancer, hepatic disease, renal disease, autoimmune disorders, diabetic retinopathy, or congenital heart disease (*n* = 182); (3) physical disability and disturbance of consciousness (*n* = 24); (4) substantial changes in dietary habits or routine activities over the previous year (*n* = 185); (5) incomplete dietary assessment (≥10% missing values) or an implausible intake of total daily energy (<700 or >4200 kcal per day for men or <500 or >3500 kcal per day for women (*n* = 11)); or (6) refusal to participate in the study (*n* = 39).

### 2.3. Ascertainment of Diseases

We defined T2D based on American Diabetes Association criteria (fasting plasma glucose > 7.0 mmol L^−1^, 2 h plasma glucose > 11.1 mmol L^−1^, or both) [26]; or medication treatment. CVDs were defined as nonfatal acute myocardial infarction, hospitalized unstable angina, and nonfatal stroke. Nonfatal myocardial infarction [27] and hospitalized unstable [28] angina were diagnosed according to the China Society of Cardiology of Chinese Medical Association criteria, including typical symptoms, elevated cardiac enzyme levels, and electrocardiographic findings. We ascertained nonfatal stroke on the basis of the national criteria, according to evidence of neurological deficits with sudden or rapid onset that persisted for a minimum of 24 h [29].

### 2.4. Data Collection

Apart from hospital documented data (e.g., clinical characteristics and clinical examinations) for cases, we used the same questionnaires to collect general and dietary factors information during the 12 months prior to diagnosis (for the cases) or an interview (for the controls). All participants were blinded to the objective of the study. In this study, both cases and controls completed a structured questionnaire via a face-to-face interview led by a well-trained dietitian. We collected information regarding (1) socio-demographic characteristics (e.g., age, sex, education level); (2) lifestyles (e.g., tobacco smoking, alcohol consumption, and tea-drinking); (3) habitual dietary consumption during the one year prior to diagnosis (for the cases) or interview (for the controls); (4) history of chronic diseases and medication use (e.g., hypertension, dyslipidemia, insulin use, and oral hypoglycemic use); and (5) physical activity. Education was grouped into three levels: primary school or below, middle, or high school, and college or above. Participants who had continuously been smoking at least one cigarette per day or drinking alcohol once per week for at least six months were defined as smokers or alcohol drinkers. We asked the participants to report their alcohol consumption using a structured questionnaire that ascertained the consumption of alcoholic beverages typical in the region (beer, wine, and Chinese spirits). The assumption for the average alcohol content (%) of beer, wine, and Chinese spirits was 4.0%, 12.5%, and 50.0%, respectively. We calculated alcohol intake (g/day) by multiplying the amount of the beverage (mL), the respective alcohol content (%), and the constant 0.80 to transform alcohol volumes into weight (g). We considered low and high intakes of alcoholic beverages as the alcohol consumption of 1–15 and more than 15 g per day, respectively. Tea drinkers were defined as individuals who drank tea at least twice a week. We also asked participants to report their previous knowledge of medical nutrition therapy for glucose control. We determined physical activity using a 19-item questionnaire by calculating the products of the time spent on a variety of activities (e.g., work, transportation, housework, leisure sedentary activity, and physical exercise) with the mean metabolic equivalent (MET) for that activity [30]. We measured participants’ height and weight while they wore only light clothes and were barefoot. We measured height and weight using standardized equipment and to the nearest 0.1 cm and 0.1 kg (measuring rod for Seca 220 column scales, SECA^®^, Hamburg, Germany), respectively. We calculated body mass index (BMI) as weight divided by height squared (kg/m^2^) [31]. We measured systolic and diastolic blood pressure (SBP and DBP, respectively) three times in 5 min time intervals using an intelligent electronic blood pressure monitor (Omron HEM-752 intelligent electronic blood pressure monitor, OMRON^®^, Japan) with an appropriate cuff size for all participants on the right arm of seated participants after a 15 min rest. We defined hypertension as patients with a mean SBP ≥ 140 mmHg and/or DBP ≥ 90 mmHg and/or as patients regularly using antihypertensive drugs [32]. We collected blood samples from the antecubital veins of the participants, which we analyzed at a central research laboratory to obtain measurements of total cholesterol, low-density lipoprotein cholesterol (LDL-C), high-density lipoprotein cholesterol (HDL-C), and triglycerides, which we determined with an automated biochemical analyzer (Beckman Coulter AU5821 automatic biochemical analyzer, BECKMAN COULTER^®^, Brea, CA, USA). Individuals whose cholesterol level was ≥200 mg/dL, LDL-C level was ≥130 mg/dL, HDL-C level was <40 mg/dL, or triglyceride level was ≥160 mg/dL or who were regularly taking lipid-lowering medications were defined as dyslipidemic according to the China Atherosclerosis Society guidelines [33].

### 2.5. Dietary Intake Assessment

We used a validated 79-item semiquantitative, paper-based, food-frequency questionnaire (FFQ) to collect dietary information [34]. For each food item, there were five possible answers (i.e., never, per year, per month, per week, and per day), and respondents provided one of two predefined amounts of food consumption (i.e., servings or portion sizes) according to their preference during the previous year. We provided photographs of generic foods and standard portion sizes to help the participants estimate the amount of food they had usually ingested. We then converted the selected choice for each food to grams per day. We transformed daily food and nutrient in-takes into standard portions according to the 2016 Dietary Guidelines for the Chinese and the 2013–2014 Food Patterns Equivalents Database, respectively [22,35]. We calculated the daily dietary intakes of nutrients and the total energy based on the Chinese Food Composition Table, 2009 [36]. Regarding salt ingestion, the FFQ does not collect detailed salt intake information, so we thus appointed a score to reflect a participants’ sodium consumption according to taste preference (i.e., very salty, salty, moderate, mild, or very mild) ranging from 0 to 10 in 2.5 increments.

### 2.6. Calculation of Diet Quality Scores

#### 2.6.1. Chinese Healthy Eating Index (CHEI)

The CHEI [22] is the first instrument available in China that can be used to assess the overall adherence to the updated Dietary Guidelines for Chinese (DGC-2016) [24]. The total score of the CHEI ranges from 0 to 100, with 100 being perfect adherence and 0 being complete nonadherence. The CHEI score is obtained on the basis of 17 components, including 12 adequacy and 5 moderation food groups. Total grains (0–5 points), whole grains and mixed beans (0–5 points), tubers (0–5 points), total vegetables (0–5 points), dark vegetables (0–5 points), dairy (0–5 points), soybeans (0–5 points), fish and seafood (0–5 points), poultry (0–5 points), eggs (0–5 points), seeds and nuts (0–5 points), and fruit (0–10 points) are the adequacy components, representing dietary elements that are encouraged, where higher scores reflect higher intake. Red meat (0–5 points), cooking oils (0–10 points), sodium (0–10 points), added sugars (0–5 points), and alcohol (0–5 points) are the moderation components, which are dietary elements whose intake is recommended to be limited, where higher scores reflect lower ingestion amounts. The intermediate intake of every component is proportionally calculated. A score is assigned to each component and a total CHEI score is generated by calculating the score for each component. Each component is adjusted for total energy using the density method (per kilocalorie), except for sugar (percentage of energy) and alcohol (absolute consumption). Higher overall CHEI scores indicate better adherence with the latest Dietary Guidelines for the Chinese, and the validity and reliability of the CHEI have been explicitly examined [37]. Details of the CHEI are listed in Appendix A.

#### 2.6.2. Healthy Eating Index (HEI)-2015

The HEI-2015 score [23], ranging from 0 to 100 possible points, was developed by the United States Department of Agriculture (USDA) to evaluate adherence to the 2015–2020 Dietary Guidelines for Americans (2015-2020 DGA) [25]. The HEI-2015 consists of 13 components, including 9 adequacy and 4 moderation components that are scored based on energy-adjusted food and nutrient intakes. The 3 adequacy components (i.e., whole grains, dairy, and fatty acids) are worth 0 to 10 points each, with 10 indicating the highest and 0 indicating the lowest consumption. The remaining 6 adequacy components (i.e., total fruits, whole fruits, total vegetables, greens and beans, total protein foods, and seafood and plant proteins) are worth 0 to 5 points each, with 5 representing the highest and 0 representing the lowest ingestion. The 4 moderation components (i.e., refined grains, sodium, added sugars, and saturated fats) are recommended to be limited, with the lowest consumption of these dietary elements scored as 10 and the highest scored as 0. Scores are proportionately calculated according to the consumption between the minimum and maximum standards [23]. A higher score indicates stricter adherence to DGA recommendations and a higher diet quality [23]. The HEI-2015 has good validity and reliability for assessing the diet quality relative to the updated Dietary Guidelines for Americans [38]. The details of the HEI-2015 are provided in Appendix A.

#### 2.6.3. Conversion of Collected Data

We divided the scores of each component of the CHEI and HEI-2015 into four groups according to the range of percentage of full scores ((individual score/full score) × 100%): 0% (0 points), 0.1–49.9%, 50–99.9%, and 100% (full points), with values of 0, 0.1–49% of maximum score, 50% of maximum score, and maximum score, respectively. For full scores of 5, the cutoffs were 0, 0.1–2.4, 2.5–4.9, and 5.0; for full scores of 10, the cutoffs were 0, 0.1–4.9, 5–9.9, and 10.0, respectively.

### 2.7. Statistical Analysis

All analyses were performed for men and women combined, except for specific analyses stratified by sex. For group comparisons, we used the independent t-test for continuous variables with a normal distribution, the Wilcoxon signed-ranks test for continuous variables with a skewed distribution, and the chi-squared test or Fisher’s exact test for the categorical variables, where appropriate.

We estimated the odds ratios (ORs) and 95% confidence intervals (CIs) for CVD using conditional logistic regression models, and we report the scores as continuous variables (in 5-point increments). We calculated the crude OR in the univariate model, and adjusted the multivariable model for age (year), BMI (kg/m^2^), marital status (married or unmarried), physical activity (MET-h/d), education level (primary education degree or below, middle or high school, or college degree or above), smoking status (yes or no), alcohol consumption (no intake, low intake, or high intake), tea-drinking status (yes or no), hypertension and dyslipidemia status (yes or no), T2D duration (year), antidiabetic medication use (yes or no), medical nutrition therapy knowledge (yes or no), and nonalcohol energy intake (kcal/d). We repeated all analyses with unconditional logistic regression modeling, because if perfect matching was not possible, performing a strict matched analysis would result in the loss of relevant information [39].

In unconditional logistic regression, we conducted stratified analyses by sex (women vs. men), BMI (≥24 vs. <24 kg/m^2^), smoking status (yes vs. no), alcohol consumption (yes vs. no), tea-drinking (yes vs. no), hypertension (yes vs. no), dyslipidemia (yes vs. no), T2D duration (≥5 vs. <5 years), medical nutrition therapy knowledge (yes or no) and, as these factors (except for sex) were not matched between case and control subjects. We further calculated the multiplicative interactions by including each interaction item in the conditional logistic regression. We conducted all analyses using SPSS 23.0 (IBM Corp., Armonk, NY, USA) and considered *p* < 0.05 statistically significant.

## 3. Results

### 3.1. Characteristics of the Participants

We included a total of 419 eligible cases and 419 T2D controls, with and without CVD, frequency-matched by sex and age, in this study. Their selected characteristics (235 and 184 pairs of men and women, respectively) are shown in Table 1. The mean ±SD ages were 62.1 ± 9.7 years in the cases and 62.1 ± 9.5 years in the controls. Compared with controls, the cases had significantly lower mean CHEI scores (65.34 ± 9.48 vs. 71.31 ± 9.05; *p* < 0.001) and HEI-2015 scores (54.03 ± 6.09 vs. 57.77 ± 6.79; *p* < 0.001). The cases had lower physical activity levels, education level, T2D duration, and proportion of tea-drinking and antidiabetic medication use, but they had a higher BMI. We found they had a higher proportion of hypertension than the controls (all *p* < 0.05). The CHEI scores significantly correlated with the HEI-2015 scores (Spearman’s *r* = 0.724; *p* < 0.001).

### 3.2. Participants in the Percentage Distribution for Each Component

As shown in Table 2, more than 50% of individuals consumed the recommended (obtained the maximum points) amounts for total grains, total vegetable, dark vegetables, fish and seafood, poultry, seeds, and nuts, added sugars, and alcohol in the CHEI, and for total vegetable, greens and beans, seafood and plant proteins, total protein foods, fatty acids, saturated fats, and added sugars in the HEI-2015. However, the intakes of whole grains and mixed beans (65.5%), dairy (71.9%), soybeans (59.8%), and fruits (78.6%) in the CHEI, and total fruits (80.7%), whole fruits (53.6%), whole grains (87.9%), dairy (91.4%), and refined grains (94.9%) in the HEI-2015 were relatively seriously deficient (did not meet 50% of the recommendations) among the components. The score differences in 10 components (whole grains and mixed beans, total vegetables, dark vegetables, tubers, dairy, soybeans, eggs, red meats, fruits, and cooking oils) of the CHEI and five components (whole grains, total vegetables, total fruits, whole fruits, and dairy) of the HEI-2015 be-tween cases and controls were all significant (all *p* < 0.05). The control group scored better than the case group in the consumption of whole grains and mixed beans, total vegetables, dark vegetables, tubers, dairy, soybeans, eggs, fruits, and cooking oils in the CHEI, and whole grains, total vegetables, total fruits, whole fruits, and dairy in the HEI-2015.

### 3.3. Total Risk Score and Stratified Analysis

The associations between the CHEI and HEI-2015 scores and CVD risk are shown in Table 3. In the univariate model, participants with higher CHEI and HEI-2015 scores had a significantly lower risk of CVD (both *p* < 0.05). In the multivariable model, the risk persisted, and the ORs (95% CIs) were 0.68 (0.61, 0.76) and 0.60 (0.52, 0.70) for per five-point increments in the CHEI and HEI-2015, respectively. In a stratified analysis of the subjects, the protective associations between the CHEI scores and CVD risk did not materially change according to the subgroups of sex, BMI, smoking status, tea-drinking, hypertension state, dyslipidemia state, T2D duration, or medical nutrition therapy knowledge (*p* for interaction ranged from 0.062 to 0.725). The relationship between HEI-2015 and CVD risk might be attenuated by the covariates of hypertension, dyslipidemia (all *p* for interaction < 0.05).

The analyses were repeated with unconditional logistic regression modeling and were not substantially different from the conditional logistic regression modeling for all comparisons (data not shown).

### 3.4. Association of Each Component Score with CVD

Figure 1 presents the ORs and 95% CIs of the score for each component of the CHEI (Figure 1A) and HEI-2015 (Figure 1B) of CVD risk between the case and control groups. Compared with the case group, a lower risk of CVD was associated with higher scores for the following foods: whole grains and mixed beans (OR, 0.84; 95% CI, 0.77, 0.91), total vegetables (OR, 0.62; 95% CI, 0.46, 0.83), dark vegetables (OR, 0.70; 95% CI, 0.57, 0.86), dairy (OR, 0.83; 95% CI, 0.76, 0.90), soybeans (OR, 0.91; 95% CI, 0.84, 0.99), eggs (OR, 0.77; 95% CI, 0.68, 0.86), fruits (OR, 0.87; 95% CI, 0.82, 0.93), cooking oils (OR, 0.87; 95% CI, 0.80, 0.95) and sodium (OR, 0.85; 95% CI, 0.77, 0.94) in the CHEI (all *p* < 0.05); and total fruits (OR, 0.76; 95% CI, 0.68, 0.86), whole fruits (OR, 0.80; 95% CI, 0.73, 0.87), total vegetables (OR, 0.46; 95% CI, 0.25, 0.86), whole grains (OR, 0.92; 95% CI, 0.87, 0.97), dairy (OR, 0.83; 95% CI, 0.76, 0.91), fatty acids (OR, 0.85; 95% CI, 0.74, 0.98), and sodium (OR, 0.85; 95% CI, 0.77, 0.94) in the HEI-2015 (all *p* < 0.05).

## 4. Discussion

In this 1:1 case–control study with 419 pairs of hospital-based CVD cases and T2D controls conducted in south China, we found that a higher score on the CHEI or HEI-2015 after diabetes diagnosis was strongly and positively associated with a lower risk of cardio-vascular outcomes among Chinese adults with diabetes.

The results of several prior studies and meta-analyses [14,15,16,40,41] have consistently shown that high diet quality, as assessed by the HEI, AHEI, DASH, and Med score, is inversely associated with the risk of CVD incidence in the general population. However, T2D patients have a higher risk of CVD, so the findings from the general population might not be directedly applicable to patients with T2D due to the potential differences between populations with and without. In previous studies, significant associations were found among diabetes patients between nut [42], and whole-grain or bran [43] consumption and a lower risk of CVDs. Moreover, the American Diabetes Association’s latest definition of a healthy diet focuses on dietary patterns rather than specific nutrients or foods [4]. These recommendations parallel the most recent 2016 Dietary Guidelines for Chinese and 2015 Dietary Guidelines for Americans, with a primary focus on moving from single-nutrient recommendations toward beneficial dietary patterns [24,25]. Significant inverse associations were reported between diet quality scores, which reflect dietary guidelines, and glycemic control or CVD risk in patients with T2D. In a 2019 cross-sectional study [44] of 229 outpatients diagnosed with T2D in Brazil, those with a lower diet quality, defined as an HEI-2010 score of <65%, had poor glycemic control. Similarly, in another cross-section study by Huffman [45], diabetes status was used as one of the covariates to predict 10-year coronary heart disease risk in Cuban Americans; the results showed that for every unit increase in the AHEI-2005 score, there was a 0.24-point reduction in the 10-year coronary heart disease risk score among participants with T2D. These results suggest that individuals with T2D may need extra support from healthcare professionals to improve their diet quality and then manage their CVD risk. However, the absence of data on CVDs incidence outcomes among T2D patients in these studies prevented direct comparisons between the various findings.

Even though we do not yet know how to prevent patients with diabetes who are at high risk of CVDs from experiencing cardiovascular events by eating properly, observational studies have reported findings consistent with ours, showing a significant protective effect of the mediating cardiovascular risk factors profile, because one of the treatment targets of nutrition therapy for patients with diabetes is to attain individualized glycemic, blood pressure, and lipid goals. For instance, the findings of a cross-sectional study conducted among 230 women with T2D showed that those with the highest dietary quality, assessed by the HEI-2010, had lower fasting blood sugar levels (148.92 ± 6.05 mg dL^−1^ vs. 171.30 ± 5.79 mg dL^−1^, *p* = 0.021), compared with the lowest tertile [46]. The findings of study involving 2,568 participants at 57 diabetes clinics, with their diet assessed with the European Prospective Investigation into Cancer and Nutrition (EPIC) questionnaire and their dietary quality evaluated with the relative Mediterranean diet score (rMED), showed that compared with a low rMED score (0–6 points), a high score (11–18 point) negatively associated with values of plasma lipids (LDL-C: 101.5 ± 31.2 vs. 105.1 ± 31.9, *p* for trend = 0.035; HDL-C: 46.8 ± 12.4 vs. 45.3 ± 11.6, *p* for trend = 0.032; triglycerides: 146.7 ± 71.0 vs. 156.2 ± 78.6, *p* for trend = 0.040), blood pressure (SBP: 133.3 ± 23.7 vs. 135.3 ± 14.9, *p* for trend = 0.045; DBP: 78.6 ± 8.5 vs. 80.7 ± 8.7, *p* for trend < 0.001), as well as glycated hemoglobin (7.63 ± 0.48 vs. 7.69 ± 0.52, *p* for trend = 0.038), suggesting that stricter adherence to the Mediterranean dietary model can be regarded as a suitable model for T2D [47]. Although different approaches are used to award optimal scores in the MED, aMED, HEI-2010, HEI-2015, and CHEI, these healthy eating diets are all characterized by a high intake of vegetables, whole grains, fruits, legumes, dairy products seeds and nuts, fish and sea-food, and lean poultry, as well as a moderate intake of alcohol and a low intake of red meat, cooking oils, sodium, and added sugars. To some extent, our observations are consistent with the existing evidence, implying that adhering to a healthy dietary pattern after a diabetes diagnosis might contribute to the prevention of cardiovascular complications among patients with T2D.

We also explored the relationships between the score for each food group on the CHEI and HEI-2015 and the achievement of treatment targets for participant risks. The beneficial effects of high-quality dietary scores may reflect the synergistic effects of diverse foods, characterized by a higher intake of vegetables, fruit, whole grains, soybeans, and dairy and a moderate intake of cooking oils and sodium. A review focused on observational and experimental studies showed that whole grains and vegetables are primary sources of dietary fiber, which might be conducive to improving dyslipidemia owing to their low glycemic index and anticholesterolemic actions [48]. Similarly, a study of 772 high-risk subjects who had either T2D or three or more of certain risk factors (current smoking, hypertension, LDL-C ≥ 160 mg/dL, HDL-C ≤ 40 mg/dL, and BMI ≥ 25 kg/m^2^) consistently showed that fasting glucose decreased but HDL-C increased with increasing dietary fiber intake (both *p* for trend <0.05). Changes in fasting glucose and TC were −13.39 mg/dL (95% CI, −19.86 to −6.93) and −9.73 mg/dL (95% CI, −17.96 to −1.49), significantly differed between the extreme quintiles of dietary fiber intake (*vs.* quintile 1) (*p* < 0.001 and 0.021, respectively) [49]. In addition, in a meta-analysis involving 14 cohort studies and five randomized control trials, sodium reduction significantly reduced resting SBP by 3.39 mm Hg (95% CI, 2.46 to 4.31) and resting DBP by 1.54 mm Hg (95% CI, 0.98 to 2.11) among adults (both *p* < 0.05). The findings showed that <2 g/day compared with ≥2 g/day of sodium intake led to a reduction in SBP by 3.47 mm Hg (95% CI, 0.76 to 6.18) and in DBP by 1.81 mm Hg (95% CI, 0.54 to 3.08) (both *p* < 0.05) [50]. Furthermore, these foods are rich sources of micronutrients including minerals, vitamins, and phytochemicals, all of which have insulin-sensitizing properties, are anti-inflammatory, and can reduce hypercoagulability, regulate metabolic and antioxidant pathways to improve macro- and microvascular status [51]. The importance of a healthy dietary pattern mainly lies in its combined effect among all types of foods and nutrients instead of on any single component. Nonetheless, because our study was an observational study, this association should be interpreted with caution, as future biological mechanical research and possible interventional studies are needed to further illustrate potential mechanisms through which cardiovascular events can be prevented among patients with T2D.

In alcohol-consumption-stratified analyses, the favorable association between CHEI and cardiovascular events remained significant in nondrinkers but not in drinkers. Considering the number of participants who drank alcohol, this result may have occurred due to the low statistical power in this subgroup. In addition, the interaction with alcohol consumption was not statistically significant, which does not indicate a different association among drinkers and nondrinkers. Additionally, we further assessed the CHEI score after removing alcohol consumption from the categories, and the results remained similar.

Of note, prior studies indicated the potential effect of BMI on the cardiovascular risk factors profile and on the adherence to a healthy diet. For example, Vitale et al. evaluated the adherence to the Mediterranean diet by the rMED, and reported that a high dietary score was associated not only with lower values of plasma lipids, blood pressure, and glucose, but also with lower BMI [47]. Similar findings were reported in a recent review, which examined 80 eligible studies, suggesting evidence of a reduction in BMI due to adherence to the Mediterranean diet, and diets characterized by a low carbohydrate, high protein, low fat consumption, and a low glycemic index load [52]. In addition, a population-based cohort study involving 79,003 women (44%) and men (56%) from the Swedish Mammography Cohort (SMC) and the Cohort of Swedish Men (COSM) (1997–2017) suggested that individuals with obesity (BMI: 30+ kg/m^2^) who strictly adhered to a Mediterranean-type diet still had higher CVD mortality, although they did not experience the increased overall mortality otherwise associated with high BMI [53]. Moreover, lower BMI did not counter the increased mortality associated with low adherence to a Mediterranean diet [53]. However, as we did not find supporting evidence of the interaction of BMI in the association between two dietary quality scores and CVD risk, further discussion of the possible effect of BMI would be helpful in interpreting the results.

In our study, we used two diet quality indices, the CHEI and the HEI-2015. Although these indices are closely correlated and share some common dietary components, they also have some notable differences. Plant-based Chinese diets are mainly rich in grains and vegetables. Compared with the HEI-2015, the CHEI emphasizes several specific dietary components that are typical in Chinese diets, such as a high intake of dark vegetables, total grains, tubers, mixed beans, soybeans, and seeds and nuts and a limited intake of cooking oil, red meat, and alcohol. All the dietary components in the CHEI are foods, whereas both foods and nutrients are considered in the HEI-2015. Moreover, researchers exploring associations between dietary patterns and risk of CVD also found slightly different results with food-based scores compared with nutrient-based scores [54]. To be consistent with international standards and to ensure comparability of our findings with those of other studies, we also introduced the HEI-2015, even though we recognized the substantial differences in the dietary patterns between Westerners and Chinese. As the most frequently studied dietary quality indices, HEI-2015, aHEI, MED, and DASH are almost all derived from the diets of Westerners, so they may not fully capture the characteristics of Chinese diets. Thus, in the present study, we also introduced the CHEI, in accordance with the latest Dietary Guidelines for Chinese.

To the best of our knowledge, we are the first to assess T2D patients’ CVD risk associated with adherence to DGC-2016 and/or 2015–2020 DGA with a case–control design. Moreover, we only included newly diagnosed CVD patients with comparable age and sex to minimize recall bias. Furthermore, we excluded the participants who had substantially changed their diet during the one year prior to the study to ensure the representativeness of the habitual diet before diagnosis or interview. Additionally, we included multiple potential covariates, including explicit risk factors of CVD (i.e., hypertension status, dyslipidemia status, and antidiabetic medication use) in the analyses to reduce the effects of residual confounding. Information bias was also further minimized because the participants were blinded to the objective of the study.

Nevertheless, several limitations of our study should be acknowledged. First, reverse causality could not be ruled out with this case–control design because we assessed dietary intake information after the diagnosis of CVD. To minimize this possibility, we only included those diagnosed with CVD within 2 weeks in our study, and obtained the dietary information in cases using the FFQ from the past year prior to diagnosis. Second, although the FFQs used in our study were validated and implemented during face-to-face interviews by well-trained dietitians, dietary measurement errors are inevitable [55]. Third, the outcomes for the sodium component should be interpreted with caution because discretionary salt being used in cooking was not accurately captured in our FFQs. As with previous epidemiological studies, this estimate is crude, and likely to underestimate the ingestion of dietary sodium. Finally, prior dietary indexes were derived on the basis of current learning, and the CHEI and HEI-2015 were originally directed toward general populations, rather than those with T2D, to prevent chronic disease. Future research is needed to develop an index adapted to T2D-relevant dietary components, such as moderation of the intake of total and refined carbohydrates, which contributes to CVD.

## 5. Conclusions

The findings from this case–control study suggest that increased adherence to either Chinese or American dietary guidelines, as reflected in the CHEI and HEI-2015, were associated with a substantially lower risk of CVD among Chinese patients with T2D. Our results further support the current recommendation that patients with diabetes adopting a healthy dietary pattern is a promising strategy for the prevention of CVD complications. Further studies, especially large prospective studies, are needed to replicate these findings.

## Figures and Tables

**Figure 1 nutrients-14-01713-f001:**
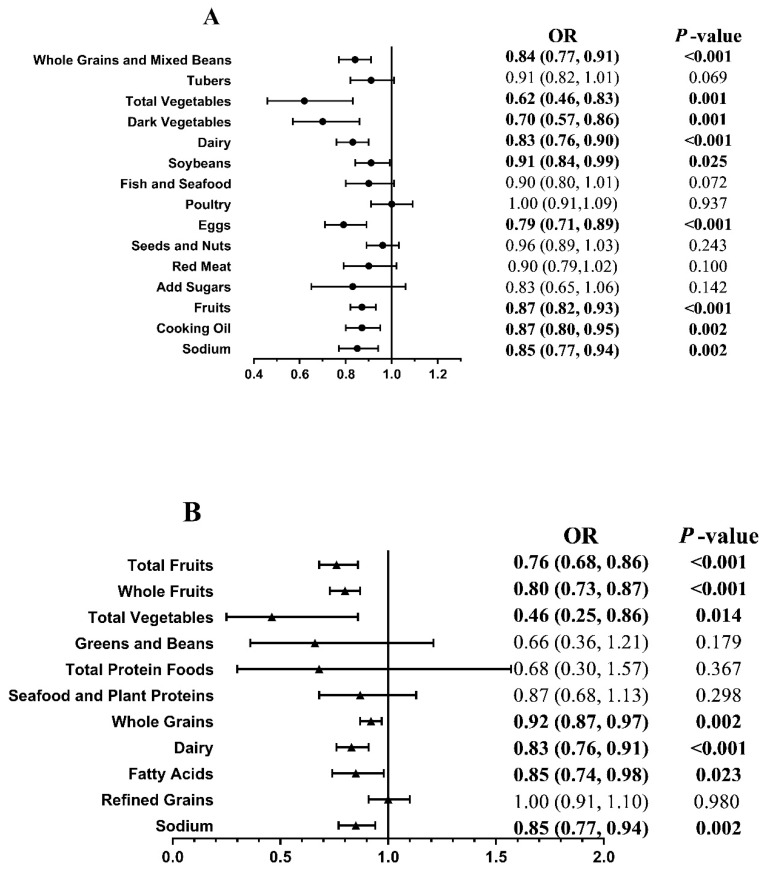
Association of the score for each component of the CHEI (**A**) and HEI-2015 (**B**) with the risk of cardiovascular complications between case and control. We adjusted ORs for age, body mass index, marital status, physical activity, education level, smoker status, alcohol consumption, tea-drinking status, hypertension status, dyslipidemia status, T2D duration, antidiabetic medication use, medical nutrition therapy knowledge, and nonalcohol energy. Cases, T2D with newly diagnosed CVD. Controls, T2D-only controls without a diagnosis of CVD. CHEI, the Chinese Healthy Eating Index. HEI-2015, the Healthy Eating Index-2015. Fatty acids, ratio of total unsaturated fatty acids (poly- and monounsaturated fatty acids (PUFAs and MUFAs)) to saturated fatty acids (SFAs). Statistically significant results are presented in bold.

**Table 1 nutrients-14-01713-t001:** Characteristics of the study participants.

Variable	Case ^a^ (*n* = 419)	Control ^b^ (*n* = 419)	*p*-Value
Age (y)	62.1 (9.7)	62.1 (9.5)	0.940
Sex, *n* (%)			--
Female	184 (43.9)	184 (43.9)	
Male	235 (56.1)	235 (56.1)	
BMI (kg/m^2^)	24.37 (3.26)	23.79 (3.47)	0.013
Smoker (%)			0.282
Yes	131 (31.3)	115 (27.4)	
No	287 (68.7)	304 (72.6)	
Alcohol consumption, *n* (%)			0.751
No Intake (0 g/d)	375 (89.5)	370 (88.3)	
Low Intake (0~15 g/d)	39 (9.3)	45 (10.7)	
High Intake (>15 g/d)	5 (1.2)	4 (1.0)	
Tea drinking, *n* (%)	188 (44.9)	220 (52.5)	0.027
Physical activity (MET-h/d) ^c^	25.56 (23.93, 27.65)	26.05 (24.38, 28.46)	0.015
Marital status, married, *n* (%)	396 (94.5)	401 (95.7)	0.423
Education level, *n* (%)			0.002
<Middle school	163 (39.4)	148 (35.4)	
Middle/High school	132 (31.9)	102 (24.4)	
≥College	119 (28.7)	168 (40.2)	
Hypertension, *n* (%)	313 (75.2)	206 (49.5)	<0.001
Dyslipidemia, *n* (%)	234 (61.7)	230 (57.1)	0.184
T2D duration (y)	7.1 (6.21)	8.96 (6.74)	<0.001
Antidiabetic medication use, *n* (%)	375 (97.9)	408 (99.5)	0.043
Medical nutrition therapy knowledge, *n* (%)	128 (30.5)	187 (44.6)	<0.001
CHEI ^d^	65.34 (9.48)	71.31 (9.05)	<0.001
HEI-2015 ^e^	54.03 (6.09)	57.77 (6.79)	<0.001
Total Energy (kcal/d) ^c,f^	1393.40 (1187.10, 1696.15)	1439.00 (1241.70, 1703.70)	0.160

^a^ T2D with newly diagnosed CVD. ^b^ T2D-only controls without a diagnosis of CVD. ^c^ Values are mean SD or median (P_25_, P_75_), where appropriate. ^d^ the Chinese Healthy Eating Index. ^e^ the Healthy Eating Index-2015. ^f^ Total energy intake was dietary energy except for alcohol.

**Table 2 nutrients-14-01713-t002:** Comparison of the percentage distribution of the cases and controls according to the scores on each item in the CHEI and HEI-2015.

Components	CHEI ^a^		*p*-Value	Components	HEI-2015 ^b^		*p*-Value
	Case ^c^	Control ^d^			Case	Control	
Total Grains *			-	Whole Grains *			<0.001
0.0% (0 point)	0 (0.0)	0 (0.0)		0.0% (0 point)	328 (78.3)	265 (63.2)	
0.1~49.9%	0 (0.0)	0 (0.0)		0.1~49.9%	57 (13.6)	87 (20.8)	
50.0~99.9%	0 (0.0)	0 (0.0)		50.0~99.9%	6 (1.4)	25 (6.0)	
100.0% (full points)	419 (100.0)	419 (100.0)		100.0% (full points)	28 (6.7)	42 (10.0)	
Whole Grains & Mixed Beans *			<0.001	Refined Grains ^#^			0.120
0.0% (0 point)	110 (26.3)	67 (16.0)		0.0% (0 point)	297 (70.9)	317 (75.6)	
0.1~49.9%	201 (48.0)	171 (40.8)		0.1~49.9%	103 (24.6)	79 (18.9)	
50.0~99.9%	32 (7.6)	55 (13.1)		50.0~99.9%	19 (4.5)	23 (5.5)	
100.0% (full points)	76 (18.1)	126 (30.1)		100.0% (full points)	0 (0.0)	0 (0.0)	
Total Vegetables ^#^			0.002	Total Vegetables ^#^			0.001
0.0% (0 point)	0 (0.0)	0 (0.0)		0.0% (0 point)	0 (0.0)	0 (0.0)	
0.1~49.9%	18 (4.3)	5 (1.2)		0.1~49.9%	6 (1.4)	1 (0.2)	
50.0~99.9%	77 (18.4)	56 (13.4)		50.0~99.9%	30 (7.2)	11 (2.6)	
100.0% (full points)	324 (77.3)	358 (85.4)		100.0% (full points)	383 (91.4)	407 (97.1)	
Dark Vegetables ^#^			<0.001	Total Fruits *			<0.001
0.0% (0 point)	2 (0.5)	0 (0.0)		0.0% (0 point)	70 (16.7)	31 (7.4)	
0.1~49.9%	26 (6.2)	8 (1.9)		0.1~49.9%	295 (70.4)	280 (66.8)	
50.0~99.9%	99 (23.6)	79 (18.9)		50.0~99.9%	37 (8.8)	76 (18.1)	
100.0% (full points)	292 (69.7)	332 (79.2)		100.0% (full points)	17 (4.1)	32 (7.6)	
Tubers ^#^			0.010	Whole Fruits *			<0.001
0.0% (0 point)	5 (1.2)	4 (1.0)		0.0% (0 point)	67 (16.0)	30 (7.2)	
0.1~49.9%	146 (34.8)	106 (25.3)		0.1~49.9%	193 (46.1)	159 (37.9)	
50.0~99.9%	140 (33.4)	144 (34.4)		50.0~99.9%	105 (25.1)	123 (29.4)	
100.0% (full points)	128 (30.5)	165 (39.4)		100.0% (full points)	54 (12.9)	107 (25.5)	
Dairy *			<0.001	Dairy ^#^			0.001
0.0% (0 point)	272 (64.9)	194 (46.3)		0.0% (0 point)	79 (18.9)	44 (10.5)	
0.1~49.9%	61 (14.6)	76 (18.1)		0.1~49.9%	313 (74.7)	330 (78.8)	
50.0~99.9%	39 (9.3)	63 (15.0)		50.0~99.9%	27 (6.4)	44 (10.5)	
100.0% (full points)	47 (11.2)	86 (20.5)		100.0% (full points)	0 (0.0)	1 (0.2)	
Soybeans *			0.001	Greens & Beans ^#^			0.113
0.0% (0 point)	55 (13.1)	51 (12.2)		0.0% (0 point)	0 (0.0)	0 (0.0)	
0.1~49.9%	221 (52.7)	174 (41.5)		0.1~49.9%	4 (1.0)	1 (0.2)	
50.0~99.9%	78 (18.6)	89 (21.2)		50.0~99.9%	9 (2.1)	3 (0.7)	
100.0% (full points)	65 (15.5)	105 (25.1)		100.0% (full points)	406 (96.9)	415 (99.0)	
Fish & Seafood *			0.493	Seafood & Plant Proteins ^#^			0.153
0.0% (0 point)	12 (2.9)	11 (2.6)		0.0% (0 point)	1 (0.2)	1 (0.2)	
0.1~49.9%	55 (13.1)	41 (9.8)		0.1~49.9%	13 (3.1)	7 (1.7)	
50.0~99.9%	96 (22.9)	101 (24.1)		50.0~99.9%	35 (8.4)	23 (5.5)	
100.0% (full points)	256 (61.1)	266 (63.5)		100.0% (full points)	370 (88.3)	388 (92.6)	
Poultry *			0.700	Total Protein Foods ^#^			0.813
0.0% (0 point)	20 (4.8)	24 (5.7)		0.0% (0 point)	0 (0.0)	0 (0.0)	
0.1~49.9%	73 (17.4)	64 (15.3)		0.1~49.9%	1 (0.2)	1 (0.2)	
50.0~99.9%	84 (20.0)	78 (18.6)		50.0~99.9%	10 (2.4)	7 (1.7)	
100.0% (full points)	242 (57.8)	253 (60.4)		100.0% (full points)	408 (97.4)	411 (98.1)	
Eggs *			0.023	Fatty Acids ^#^			0.074
0.0% (0 point)	16 (3.8)	9 (2.1)		0.0% (0 point)	4 (0.01)	5 (0.01)	
0.1~49.9%	147 (35.1)	119 (28.4)		0.1~49.9%	161 (0.38)	162 (0.39)	
50.0~99.9%	173 (41.3)	178 (42.5)		50.0~99.9%	235 (0.56)	215 (0.51)	
100.0% (full points)	83 (19.8)	113 (27.0)		100.0% (full points)	19 (0.05)	37 (0.09)	
Seeds and Nuts *			0.172	Saturated Fats *			-
0.0% (0 point)	111 (26.5)	95 (22.7)		0.0% (0 point)	0 (0.0)	0 (0.0)	
0.1~49.9%	50 (11.9)	39 (9.3)		0.1~49.9%	0 (0.0)	0 (0.0)	
50.0~99.9%	15 (3.6)	23 (5.5)		50.0~99.9%	0 (0.0)	0 (0.0)	
100.0% (full points)	243 (58.0)	262 (62.5)		100.0% (full points)	419 (100.0)	419 (100.0)	
Red Meat *			0.030	Added Sugars *			-
0.0% (0 point)	32 (7.6)	13 (3.1)		0.0% (0 point)	0 (0.0)	0 (0.0)	
0.1~49.9%	93 (22.2)	106 (25.3)		0.1~49.9%	0 (0.0)	0 (0.0)	
50.0~99.9%	279 (66.6)	284 (67.8)		50.0~99.9%	0 (0.0)	0 (0.0)	
100.0% (full points)	15 (3.6)	16 (3.8)		100.0% (full points)	419 (100.0)	419 (100.0)	
Added Sugars ^#^			0.902	Sodium *			0.111
0.0% (0 point)	2 (0.5)	2 (0.5)		0.0% (0 point)	5 (1.2)	1 (0.2)	
0.1~49.9%	1 (0.2)	2 (0.5)		0.1~49.9%	52 (12.4)	47 (11.2)	
50.0~99.9%	78 (18.6)	72 (17.2)		50.0~99.9%	359 (85.7)	371 (88.5)	
100.0% (full points)	338 (80.7)	343 (81.9)		100.0% (full points)	3 (0.7)	0 (0.0)	
Alcohol *			0.124				
0.0% (0 point)	1 (0.2)	0 (0.0)					
0.1~49.9%	0 (0.0)	0 (0.0)					
50.0~99.9%	3 (0.7)	0 (0.0)					
100.0% (full points)	415 (99.0)	419 (100.0)					
Fruits *			<0.001				
0.0% (0 point)	68 (16.2)	29 (6.9)					
0.1~49.9%	290 (69.2)	272 (64.9)					
50.0~99.9%	43 (10.3)	84 (20.0)					
100.0% (full points)	18 (4.3)	34 (8.1)					
Cooking Oils ^#^			0.002				
0.0% (0 point)	4 (1.0)	2 (0.5)					
0.1~49.9%	44 (10.5)	20 (4.8)					
50.0~99.9%	225 (53.7)	214 (51.1)					
100.0% (full points)	146 (34.8)	183 (43.7)					
Sodium ^#^			0.111				
0.0% (0 point)	5 (1.2)	1 (0.2)					
0.1~49.9%	52 (12.4)	47 (11.2)					
50.0~99.9%	359 (85.7)	371 (88.5)					
100.0% (full points)	3 (0.7)	0 (0.0)					

For fruits, sodium and cooking oils in the CHEI and whole grains, dairy, fatty acids, refined grains, sodium, added sugars and saturated fats in the HEI-2015, the cutoffs for the four groups are 0.0, 0.1–4.9, 5.0–9.9 and 10.0. For the remaining components in the CHEI and the HEI-2015, the cutoffs for the four groups are 0.0, 0.1–2.4, 2.5–4.9 and 5.0. ^a^ the Chinese Healthy Eating Index. ^b^ the Healthy Eating Index-2015. ^c^ T2D with newly diagnosed CVD. ^d^ T2D-only controls without a diagnosis of CVD. *: *p*-value calculated by the chi-square test; ^#^
*p*-value calculated by the Fisher’s exact test.

**Table 3 nutrients-14-01713-t003:** ORs (95% CIs) of CVD for per 5-point increments in CHEI and HEI-2015 stratified by selected factors.

	*n* (Cases ^a^/Controls ^b^)	Crude OR (95% CI)	Multivariable-Adjusted OR (95% CI) ^c^	*p*-Interaction
**CHEI** ^d^				
**Total scored**	419/419	0.65 (0.59, 0.72)	0.68 (0.61, 0.76)	
**Sex**				0.175
Female	184/184	0.78 (0.70, 0.88)	0.84 (0.73, 0.96)	
Male	235/235	0.63 (0.56, 0.70)	0.66 (0.59, 0.75)	
**BMI, kg/m^2^**				0.435
≥24	198/184	0.76 (0.68, 0.86)	0.80 (0.71, 0.91)	
<24	221/235	0.67 (0.60, 0.75)	0.68 (0.60, 0.77)	
**Smoker**				0.436
Yes	131/115	0.62 (0.53, 0.73)	0.68 (0.57, 0.81)	
No	287/304	0.74 (0.67, 0.81)	0.76 (0.69, 0.84)	
**Alcohol consumption**				0.257
Yes	44/49	0.79 (0.61, 1.04)	0.85 (0.58, 1.24)	
No	375/370	0.70 (0.64, 0.76)	0.73 (0.66, 0.79)	
**Tea-drinking**				0.674
Yes	188/220	0.73 (0.65, 0.81)	0.71 (0.63, 0.81)	
No	231/199	0.68 (0.61, 0.77)	0.76 (0.67, 0.85)	
**Hypertension**				0.062
Yes	313/206	0.75 (0.68, 0.83)	0.78 (0.70, 0.87)	
No	103/210	0.67 (0.58, 0.77)	0.67 (0.57, 0.78)	
**Dyslipidemia**				0.725
Yes	234/230	0.76 (0.69, 0.85)	0.78 (0.70, 0.87)	
No	145/173	0.64 (0.55, 0.74)	0.68 (0.58, 0.80)	
**T2D duration, y**				0.626
≥5	227/269	0.74 (0.67, 0.82)	0.77 (0.69, 0.86)	
<5	192/150	0.66 (0.58, 0.75)	0.67 (0.58, 0.77)	
**Medical nutrition therapy knowledge**				0.301
Yes	128/187	0.79 (0.70, 0.90)	0.76 (0.65, 0.88)	
No	254/223	0.70 (0.63, 0.78)	0.71 (0.63, 0.80)	
**HEI-2015** ^e^				
**Total scored**	419/419	0.58 (0.50, 0.66)	0.60 (0.52, 0.70)	
**Sex**				0.079
Female	184/184	0.70 (0.59, 0.83)	0.74 (0.62, 0.90)	
Male	235/235	0.52 (0.43, 0.63)	0.55 (0.45, 0.67)	
**BMI, kg/m^2^**				0.242
≥24	198/184	0.70 (0.59, 0.83)	0.75 (0.62, 0.90)	
<24	221/235	0.58 (0.49, 0.68)	0.60 (0.50, 0.71)	
**Smoker**				0.504
Yes	131/115	0.51 (0.40, 0.67)	0.56 (0.42, 0.75)	
No	287/304	0.67 (0.58, 0.77)	0.68 (0.59, 0.78)	
**Alcohol consumption**				0.908
Yes	44/49	0.55 (0.36, 0.86)	0.52 (0.28, 0.98)	
No	375/370	0.63 (0.56, 0.71)	0.65 (0.57, 0.74)	
**Tea-drinking**				0.726
Yes	188/220	0.63 (0.53, 0.74)	0.65 (0.54, 0.78)	
No	231/199	0.63 (0.53, 0.74)	0.65 (0.54, 0.77)	
**Hypertension**				0.003
Yes	313/206	0.70 (0.60, 0.81)	0.72 (0.61, 0.85)	
No	103/210	0.60 (0.49, 0.74)	0.60 (0.48, 0.75)	
**Dyslipidemia**				0.031
Yes	234/230	0.70 (0.60, 0.81)	0.71 (0.60, 0.83)	
No	145/173	0.59 (0.49, 0.72)	0.64 (0.51, 0.79)	
**T2D duration, y**				0.182
≥5	227/269	0.72 (0.62, 0.84)	0.74 (0.62, 0.88)	
<5	192/150	0.53 (0.44, 0.64)	0.53 (0.43, 0.65)	
**Medical nutrition therapy knowledge**				0.169
Yes	128/187	0.82 (0.69, 0.97)	0.81 (0.67, 0.98)	
No	254/223	0.57 (0.49, 0.68)	0.56 (0.47, 0.68)	

^a^ T2D with newly diagnosed CVD. ^b^ T2D-only controls without a diagnosis of CVD. ^c^ Adjusted for age, BMI, marital status, physical activity, education level, smoking status, alcohol consumption, tea-drinking status, hypertension status, dyslipidemia status, T2D duration, antidiabetic medication use, medical nutrition therapy knowledge and non-alcohol energy intake. ^d^ the Chinese Healthy Eating Index. ^e^ the Healthy Eating Index-2015.

## Data Availability

The data presented in this study are available on request from the corresponding author. The data are not publicly available because they were not collected originally for researches.

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
