# Peer review of "Greater Adherence to Dietary Guidelines Associated with Reduced Risk of Cardiovascular Diseases in Chinese Patients with Type 2 Diabetes"

_nutrients, 2022, doi:10.3390/nu14091713_

Round 1

Reviewer 1 Report

The document has minor language corrections, it should be revised carefully by a native English speaking individual or by a English language editing service.

You should not use the terms "obese or diabetic" to refer to individuals with obesity or diabetes, as these terms tag the patients and this disease are not inherent to them. 

Author Response

The document has minor language corrections, it should be revised carefully by a native English speaking individual or by a English language editing service.

Response: Thank you for your suggestion, we apologize for the poor language of our manuscript. We worked on the manuscript for a long time and the repeated addition and removal of sentences and sections obviously led to poor readability. We have now worked on both language and readability and have also been revised by the English language editing service of MDPI. We really hope that the flow and language level have been substantially improved.

You should not use the terms "obese or diabetic" to refer to individuals with obesity or diabetes, as these terms tag the patients and this disease are not inherent to them.

Response: Thank you so much for your nice advice, and we have revised it according to your suggestion in the revised manuscript as follow:

Line 413: “…, T2D patients have a higher risk of CVD, so the findings from the general population might not be directedly applicable to patients with T2D due to the potential differences between populations with and without.”

Line 440: “Even though we do not yet know how to prevent patients with diabetes who are at high risk of CVDs from experiencing cardiovascular events by eating properly …”

Line 531: “In addition, a population-based cohort study involving 79,003 women (44%) and men (56%) from the Swedish Mammography Cohort (SMC) and the Cohort of Swedish Men (COSM) (1997-2017) suggested that individuals with obesity (BMI: 30+ kg/m2) …”

Reviewer 2 Report

Overall I found this article to be excellent and very useful in dietetic practice in T2DB. I only have a few minor points:

Line 104: It will be useful to mention the main differences between the two indices.

line 166-167: mentions that both case and controls were interviewed by a dietitian. However in line 171 it only mentions an interview for controls....this section is not clear.

line 170: What was the reason for controlling for tea drinking? Should coffee not also have been included?

Line 197: provide detail on anthropometry, equipment, models, calibration with references also a reference is required for BMI.

Line 253: the word exploited should be replaced by "developed".

Author Response

Overall I found this article to be excellent and very useful in dietetic practice in T2DB. I only have a few minor points: 

Line 104: It will be useful to mention the main differences between the two indices.

Response: Thank you so much for your valuable suggestion. We have stated this section as below:

“As such, in the current study, we thus investigated the associations between two diet quality indexes, the Chinese Healthy Eating Index (CHEI) [22] and the latest version of the HEI (HEI-2015) [23], (the food-based and food-nutrient-based indices that reflect the 2016 Dietary Guidelines for the Chinese population [24] and the 2015–2020 Dietary Guidelines for the American population [25], respectively), with the risk of CVD among patients with T2D participating in a 1:1 matched case–control study in south China. We provide some additional information for the development of dietary guide-lines for the management of T2D.”

Line 166-167: mentions that both case and controls were interviewed by a dietitian. However in line 171 it only mentions an interview for controls....this section is not clear.

Response: We feel sincere sorry for our confusing statement. In fact, both cases and controls completed the structured questionnaire via a face-to-face interviewed by a well-trained dietitian. As for habitual dietary consumption, the diagnosis of CVD might affect patients’ dietary habits, and we thus collected the one year prior to diagnosis (for the cases) or interview (for the controls). To minimize the possibility of reverse causality, we only included those diagnosed with CVD within 2 weeks in our study, and obtained the dietary information in cases using the FFQ from the past year prior to diagnosis.

Line 170: What was the reason for controlling for tea drinking? Should coffee not also have been included?

Response: Thank you. In fact, tea consumption has been reported to be associated with a lower risk of diabetes and CVD, especially comparing high-frequent consumption (3-4 cups/day) with none (Ref: Mozaffarian D: Dietary and Policy Priorities for Cardiovascular Disease, Diabetes, and Obesity: A Comprehensive Review. Circulation 2016;133:187-225). Consider that drinking tea has become an important part of daily life among Chinese people (the prevalence being 48.7% in our study), tea-drinking status has been included into the multivariable model, though plausible biologic mechanisms that could explain the association between tea and diabetes risk have not been confirmed. As for coffee, the consumption in our population is low (the prevalence being 3.9% in our study), and the average drinking volume is only 66 ml per day in coffee-drinking participants.

To control for confounding by coffee drinking, a sensitivity analysis further adjusted coffee-drinking status (yes or no) were calculated in the study participants, and it also did not change our conclusion.

n

(cases/ controls)

Multivariable- adjusted OR (95% CI) a

CHEI

419/419

0.68(0.61,0.76)

HEI-2015

419/419

0.60(0.52,0.70)

Abbreviations: CHEI, Chinese Healthy Eating Index; HEI-2015, Healthy Eating Index 2015; ORs, Odds ratios, CI, Confidence Interval.

a Adjusted for age, BMI, marital status, physical activity, education level, smoking status, alcohol consumption, tea-drinking status, coffee-drinking status, hypertension status, dyslipidemia status, T2D duration, antidiabetic medication use, medical nutrition therapy knowledge and non-alcohol energy intake.

Line 197: provide detail on anthropometry, equipment, models, calibration with references also a reference is required for BMI. 

Response: Thank you so much for your valuable suggestion. We have revised the statements in the revised manuscript as below:

“We measured participants’ height and weight while they wore only light clothes and were barefoot. We measured height and weight using standardized equipment and to the nearest 0.1 cm and 0.1 kg (measuring rod for Seca 220 column scales, SECA®, Germany), respectively. We calculated body mass index (BMI) as weight divided by height squared (kg/m2) [31]. We measured systolic and diastolic blood pressure (SBP and DBP, respectively) three times in 5 min time intervals using an intelligent electronic blood pressure monitor (Omron HEM-752 intelligent electronic blood pressure monitor, OMRON®, Japan) with an appropriate cuff size for all participants on the right arm of seated participants after a 15 min rest. We defined hypertension as patients with a mean SBP ≥140 mmHg and/or DBP ≥90 mmHg and/or as patients regularly using antihypertensive drugs [32]. We collected blood samples from the antecubital veins of the participants, which we analyzed at a central research laboratory to obtain measurements of total cholesterol, low-density lipoprotein cholesterol (LDL-C), high-density lipoprotein cholesterol (HDL-C), and triglycerides, which we determined with an automated biochemical analyzer (Beckman Coulter AU5821 automatic biochemical analyzer, BECKMAN COULTER®, USA).”

Line 253: the word exploited should be replaced by "developed".

Response: Thank you so much for your nice advice, and we have revised it according to your suggestion.

“The HEI-2015 score [23], ranging from 0 to 100 possible points, was developed by the United States Department of Agriculture (USDA to evaluate adherence to the 2015–2020 Dietary Guidelines for Americans (2015-2020 DGA).”

This manuscript is a resubmission of an earlier submission. The following is a list of the peer review reports and author responses from that submission.